# Simultaneous Worlds: Supporting Fluid Exploration of Multiple Data Sets via Physical Models

Carmen Hull*
University of Calgary

Søren Knudsen†
IT University of Copenhagen

Sheelagh Carpendale‡
Simon Fraser University

Wesley Willett§
University of Calgary

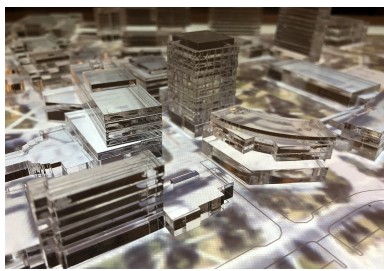 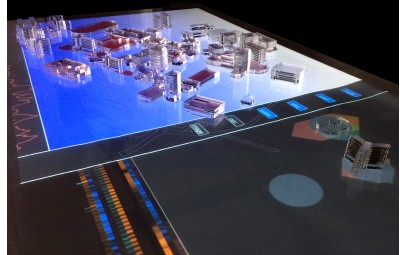 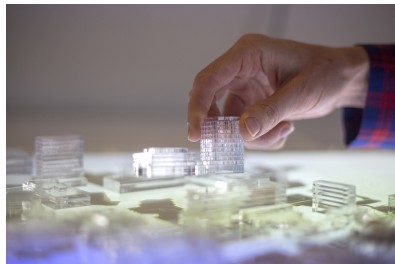

Figure 1: Examples of three opportunities for integrating visualizations and architectural scale models on tabletops. (Left) Satellite imagery shown **situated** with a physical model. (Center) Multiple data visualizations **composed** together using the geometry and position of a model. (Right) Individual buildings from a model are used to **manipulate and author** visualizations.

## ABSTRACT

We take the well-established use of physical scale models in architecture and identify new opportunities for using them to interactively visualize and examine multiple streams of geospatial data. Overlaying, comparing, or integrating visualizations of complementary data sets in the same physical space is often challenging given the constraints of various data types and the limited design space of possible visual encodings. Our vision of "simultaneous worlds" uses physical models as a substrate upon which visualizations of multiple data streams can be dynamically and concurrently integrated. To explore the potential of this concept, we created three design explorations that use an illuminated campus model to integrate visualizations about building energy use, climate, and movement paths on a university campus. We use a research through design approach, documenting how our interdisciplinary collaborations with domain experts, students, and architects informed our designs. Based on our observations, we characterize the benefits of models for 1) situating visualizations, 2) composing visualizations, and 3) manipulating and authoring visualizations. Our work highlights the potential of physical models to support embodied exploration of spatial and non-spatial visualizations through fluid interactions.

**Keywords:** Information visualization, interactive surfaces, data physicalization, architectural models

**Index Terms:** Human-centered computing—Visualization—Visualization techniques—Treemaps; Human-centered computing—Visualization—Visualization design and evaluation methods

## 1 INTRODUCTION

Although data sets are often examined in isolation, they are rarely generated that way. Rather, every piece of data represents one small element in a larger picture and captures only one of many perspectives of the places, people, and phenomena it seeks to characterize.

---

*e-mail: carmenhullstudio@gmail.com

†e-mail: soekn@itu.dk

‡e-mail: sheelagh@sfu.ca

§e-mail: wj@wjwillett.net

Overlaying, comparing, or integrating visualizations of multiple complementary data sets in the same physical space is often challenging [8], given the unique constraints of various data types and the limited design space of possible visual encodings. Moreover, for data sets that reference the physical world, much of the surrounding context remains unrecorded, and can be appreciated only by visualizing the data in-situ, where physical and temporal scales can make observation difficult. For example, it is impossible to simultaneously experience summer and winter climate conditions at the same location. Similarly, in the physical world, it is impossible to observe large scale systems, such as an entire campus or urban area, directly. As a result, it is difficult for viewers to examine many data sets at once, and viewers often miss out on ambient and environmental data that might provide context and support interpretation.

Our work proposes the concept of "simultaneous worlds", (Figure 2) which highlights how physical architectural models can provide context for and support transitions between multiple data visualizations. To explore the potential of this concept, we built a tangible table-top system using scale models of a university campus. Our tabletop system juxtaposes visualizations of operational data such as heating and cooling costs alongside ambient and contextual data sets including environmental conditions, occupancy and movement logs, and historical aerial photos. The interactive table uses rear-projection to overlay visualizations of this data with transparent trackable architectural models.

We explored this particular system with several sets of stakeholders. These included campus energy managers (who were interested in contextualizing data about energy use and weather), architects (who were interested in understanding patterns of human movement on campus), and students (for whom these kinds of physical models could increase awareness around important topics like energy use). In our explorations, we wanted to emphasize the broad utility of our tabletop system for use with other data visualizations including human movement and occupancy data.

For our first contribution, we examine three avenues via which physical architectural models can support data exploration and showcase the benefits they provide (Figure 1). We explore how architectural models can situate data, improving viewers' ability to identify locations and connect data to them. We then highlight how visualization developers can use models to anchor composite visualizations that combine multiple datasets and visualizations together in the same space. Finally, we show how physical models can support

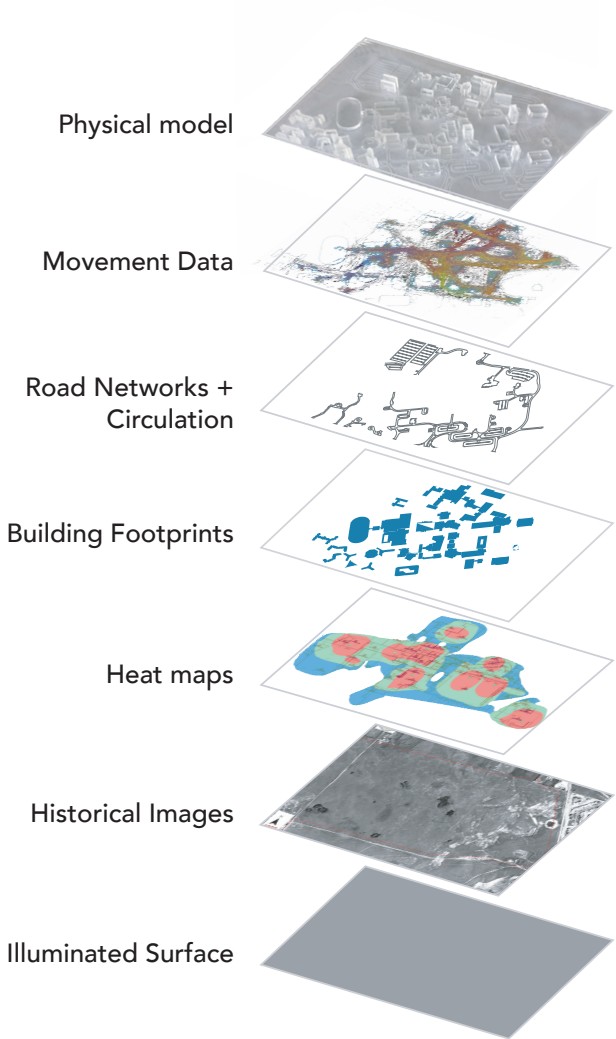

**Physical model**

**Movement Data**

**Road Networks + Circulation**

**Building Footprints**

**Heat maps**

**Historical Images**

**Illuminated Surface**

Figure 2: Multiple geospatial and visualization layers can all be visualized in the context of the same physical architectural model. These layers serve as "Simultaneous Worlds", supporting integrated exploration and reasoning.

fluid, tangible interactions which allow viewers to explore and reconfigure spatial visualizations. We then illustrate these concepts via two example data analysis tools, one of which uses our system to visualize campus climate and energy use, and one visualizing human movement across the university.

Our second contribution is the documentation of our design process using a *research through design* approach. We conducted this research as an iterative design-oriented exploration of the potential of simultaneous worlds. We collected reflections from a variety of stakeholders, including campus architects and energy managers, who participated in the design of the system. Throughout the process, we collected reflections, framings, and opportunities, using these qualitative and observational practices to guide our research work—resulting in a set of framings and prototypes that more deeply illustrate the potential of architectural models to serve as tangible and context-specific interfaces for data visualizations.

Our initial findings show that the models provide immediate and familiar symbols that allow the user to quickly understand visual encodings in a variety of different visualizations without annotation or lengthy explanation, and provide additional benefits related

to the geometric and spatial characteristics of the model. We conclude with a discussion on additional possible application areas and considerations for applying the concept of simultaneous worlds for visualization researchers and designers of tabletop systems.

## 2 RELATED WORK

Traditionally, most data visualization tools have focused on creating new visual representations that support the intentional exploration of specific data sets of interest. Yet, in practice, interpreting data and making informed decisions often calls for additional context—which situates the data with respect to locations, events, and phenomena not captured in the data itself. To address this, we explore how physical models can serve as a substrate for data analysis tasks, providing a common set of anchors upon which to display both operational data that drives analyses and ambient data which provides context to them. Our work sits at the intersection of research on physical architectural models, situated data visualizations, and tangible interfaces.

### 2.1 Architectural Models and Tabletops

Digitally-augmented physical and architectural models have a relatively long history in HCI, including examples like the Metadesk [29] and URP [30] which provide some of the earliest demonstrations of the value of tangible computing. A diverse range of subsequent projects have also explored how physical modeling [2, 22], shape-changing displays [10], and augmented architectural models [28] can support physical planning, drive social engagement, and present data specific to urban settings. The classic tabletop literature has established the collaborative advantages of physical tabletop systems, allowing for shared ownership of the territory of the work space, as well as ease of use in navigation, locomotion, and turn-taking [26].

Although the current trend in urban analytics focuses on the exploration of 3D digital models in virtual reality, physical models provide an immersive experience of data within the context of a "real-world" environment that doesn't rely on VR equipment. Chandler et al. characterize the benefits of analysing urban data within a 3D model over 2D maps [19, Chapter 11] but also note some of the challenges associated with supporting collaborative discussion in virtual environments. Physical models may provide useful alternatives to these tools in a variety of application areas—including maps for emergency response, real estate development, and neighborhood planning which could leverage the collaborative benefit of tabletop models with site-specific data visualizations.

### 2.2 Situated Visualizations

Work on situated visualizations (visualizations displayed in related environments [32]) and embedded visualizations (visualizations deeply connected to specific spaces, objects, and entities [33]) highlight how visualizing data in the physical world can help provide environmental and ambient context like weather and traffic conditions. Mobile and augmented reality visualization tools [31, 32], which overlay data on top of physical referents in a viewer's surroundings represent one popular approach. Viewers in physical spaces can also observe environmental traces like paths, physical wear, and decay—and these traces give a sense of ongoing ambient processes—or create indexical visualizations [21] and Autographic Visualizations [20] that expressly illustrate ambient data in the environment.

However, in many cases, the distance, size, or physical inaccessibility of relevant environments can make it difficult or impossible to display data on top of them to support in-situ analysis and decision-making. Moreover, ambient data that could provide context about spaces and phenomena may not be visible to the naked eye and may span larger timescales or geographic extents than a viewer can reasonably observe. In response, we examine how architectural scale models [5] can serve as facsimiles or proxies for real-world environments [33], providing anchors upon which both operational and ambient data can be examined and integrated.

## 2.3 Tangibles on Tabletops

According to Ishii, "the key idea of Tangible User Interfaces (TUIs) is to give physical forms to digital information...to serve as both representation and controls for their digital counterparts" [14]. Many projects embody either representation or control, but not both. Most often, tangibles are used as tools for interaction and control, such as TZee objects [34] and Lumino [3]. Other projects in architecture and urban planning also consider tangibles. The MIT CityScope uses projection onto Lego objects for urban planning and other scenarios [2], while Maquil et al.'s ColorTable [18] uses simple primitive forms for both representing generic road and wall forms and as input devices. The generic forms used in ColorTable, however, do not show important details such as height, context, or real-world scale—three variables identified in immersive analytics [19] as essential for urban design analysis. Within most of the existing tangible tabletop projects, there is a missed opportunity to encode meaning in the material and geometric properties of the object. More recent systems like Ens et al.'s Uplift [8], meanwhile, have mostly focused on physical models as a background for augmented reality visualizations above the tabletop. Ren and Heiecker further support the use of physical models over VR experiences in their 2021 study that revealed faster, more confident answers and long term memorability with physical models for data visualization [24]. Our approach is to use site-specific architectural models to display and contain the visualizations of a specific place. Like other projects, we track the models to allow people to use them as interaction handles, and by doing so, control aspects of the displayed visualizations.

From a technical perspective, occlusion is a significant problem with top down projection systems, not only because the arms of the user block the projected images, but also because the tangibles occlude whatever is on the illuminated surface beneath them. Most TUIs use visible markers for detection by a computer vision system, which requires opaque objects and top mounted projection. Tangible 3D Tabletops by Dalsgaard et al. [7] uses two projectors to project images onto 3D cubes to represent buildings, plus a bottom projector to project visuals below. Using this system, the designer can project architectural details onto the sides of the blocks, however image quality is limited by the resolution of the projector and the size of the cubes.

## 2.4 Data Physicalization

Data Physicalization [15] is an emerging research area that studies the use of material and geometric encodings to capture data. While this is a closely related area, we do not consider this work a data physicalization project as our simultaneous worlds prototypes never encode data using the physical form or properties of the model. Instead, the data visualizations remain strictly 2D while the physical models provide context, define the shape of the visualizations, and serve as interactive handles for them.

## 3 SIMULTANEOUS WORLDS

We introduce the concept of "simultaneous worlds" in which architectural models and data visualizations inhabit the same physical space. Using a *research through design* framework [35], we document our iterative design process for a 3D interactive campus model. Based on ongoing conversations with energy, building, and operations managers, as well as students and architects over the course of approximately two years, we built and revised two interactive model systems (Figure 5). We also demonstrated the system publicly seven times during its development, including as part of a citywide art and science festival, at department and educational showcase events, and lab demo days. In all cases, our intent was to examine the ability of the model to facilitate an understanding of the data more quickly, and to expand the possibilities of connections between energy use and their own experiences on campus, whether as a student or an administrator. Based on our observations, the paper seeks to highlight interesting potential areas of opportunity for integrating architectural models and visualizations. Through this lens, we illustrate how "simultaneous worlds" offers opportunities for situating spatial and non-spatial datasets and supporting complex reasoning about real world spaces.

Our work illustrates the potential for even tighter integration between data visualization and more complex architectural models than URP [30] and CityScope [2], which highlighted the potential for using simple building shapes to serve as a canvas for an interface to data and simulations. In particular, we highlight how translucent architectural models can provide a substrate for compositing visualizations of multiple complementary datasets including climate information, building automation logs, and human movement traces on tabletop displays. While each distinct information model or representation can exist on its own, we demonstrate how the physical geometry of the models can help connect related visualizations in an integrated fashion. We conducted three design explorations that examine the potential for physical architectural models to help situate, compose, and support interaction with data visualizations.

## 3.1 Tabletop Implementation and Setup

We explored these concepts in the context of a bottom-projected tabletop which we built to accommodate a 26×46 inch 1:1700 scale architectural model of a 2.13 km$^2$ university campus. This model provided a platform on which to visualize a wide variety of readily-available environmental, social, and infrastructure-related datasets.

## 3.2 Acrylic Campus Model

The physical table consists of a projector, a laminated screen with a base map etched into the surface, and a frame made of 80/20 building materials. Our system (Figure 3-left and middle) uses an acrylic model placed on the illuminated tabletop which displays a variety of different spatial visualizations. We constructed the scale model using a mix of digital fabrication and hand-building techniques. The

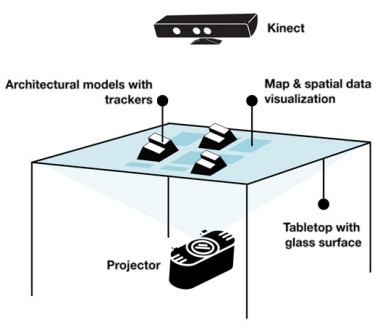 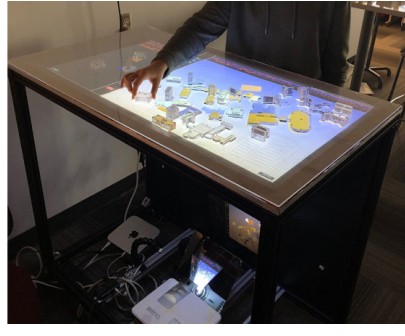 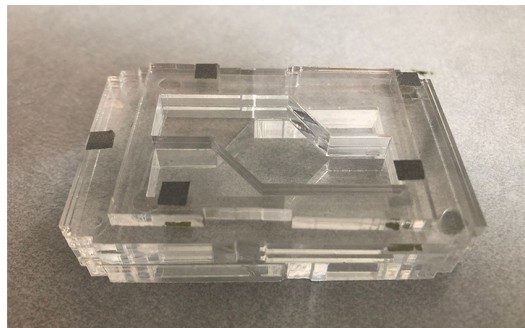

Figure 3: Detail images showing system diagram (left), photo of table (middle), and close-up of building tracker markers (right).

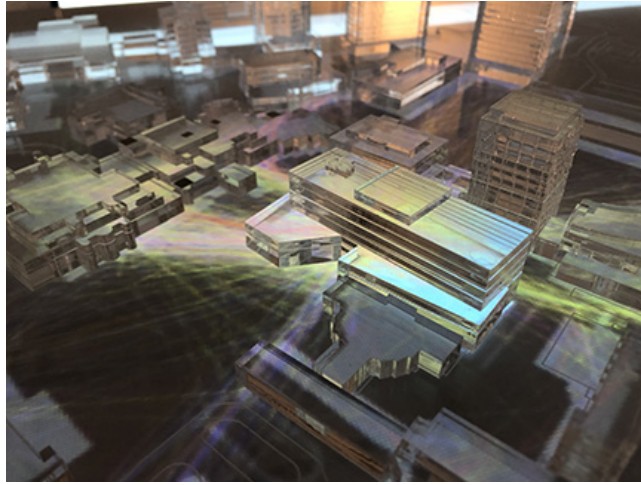

Figure 4: Scale model with our campus movement visualization.

| Design Phase | 1st Prototype (4 months) | 2nd Prototype (12 months) | Reflection (8 months) |
|---|---|---|---|
| Knowledge Production | define discover construct | sythesize refine re-construct | assess reflect sythesize |
| Collaborators | Energy Manager Facilities Director Sustainability Manager | Energy Manager Operations Manager Sustainability Director | Energy Manager 4 Architects 16 students 7 public demos |

Figure 5: Design phases and knowledge gathered through collaborations in each phase.

unique outline of every floor of each of the buildings was laser cut from a 1/8 inch acrylic sheet (which, at this scale, was roughly equivalent to the height of one floor). We then stacked and glued these layers together with a clear adhesive. We also etched the surface of the tabletop to include the footprints of each building along with roads, parking lots, trails, and other important physical elements of campus architecture. The tabletop is bottom-projected, with the visualizations visible through and around the model. Due to the translucent and internally reflective nature of the acrylic, the visualizations displayed on the surface reflect up through the building masses, filling the volumes with color.

### 3.3 Touch Surface and Tangible Interaction

To track the position of buildings on the tabletop, we developed a custom tracking system using a single Microsoft Kinect V2 and OpenCV 3 on the Unity game engine. The approach is similar to motion tracking systems like the Vicon or OptiTrack. We attach between three and seven small retro-reflective stickers to each of the buildings as tracking markers (Figure 3-right), then illuminate and track them using a Kinect mounted immediately above the tabletop. We use k-means clustering in OpenCV to group the marker positions detected by the Kinect. We then use OpenCV's machine learning tools to train a recognizer to identify buildings based on their total number of markers, the positions of the markers on their perimeters, and the compactness of the cluster. This process estimates the total number of tracked buildings on the table, and outputs positions and ids for individual recognized buildings. The system broadcasts update events via WebSockets whenever a building is placed on the table, removed from the table, or changes position.

### 3.4 Visualizations

We implemented two visualization systems for the model. The first (Figure 1-center) is an **energy use** visualization, developed using Processing, which combines building automation logs and ambient weather data from the university campus collected over a two year period. The second **movement** visualization (Figure 4), developed using HTML, Javascript, and Mapbox GL, showcases location data from several hundred university students collected between 2013 and 2017. We describe both visualizations in more detail later in the paper.

### 4 COLLABORATIONS

Throughout the design process of the tabletop model, we systematically consulted with domain experts including campus operations and energy managers at major project milestones (Figure 5). After our initial Design Phase we conducted three iterations (1st Prototype, 2nd Prototype, and Reflection) in which we collected feedback from these experts via periodic semi-structured interviews and informal demo sessions and used that input to assess and evolve the platform. After the final system was complete, we also invited four architects to reflect on the system, and discussed the impact of using physical architectural models for visualizing campus-specific data. We also demonstrated the system publicly seven times during its development, including as part of a citywide art and science festival, at department and educational showcase events, and during lab demo days.

### 4.1 Collaborations with Domain Experts

We consulted repeatedly with domain experts including the university's energy manager, operations managers, and personnel from the office of sustainability. We met with a total of five experts over two years, including multiple iterations with the energy manager and sustainability staff. These stakeholders provided access to initial raw data as well as consultation and feedback on the project, helping us to tailor the visualization design to their requirements.

In addition to this ongoing engagement, we held informal debriefing meetings near the end of the development process with members of the energy management team to collect additional feedback. We began each semi-structured interview with a demo of the current features and possible interactions, and collected responses and interactions of the participants through notes and video.

Our first interview with the campus energy manager was particularly influential, providing a deeper understanding of the campus's existing methods of energy data analysis and what the managers were looking for in a new visualization system for energy use data. The campus's existing web-based dashboard did not engage users or receive as much traffic as the team had hoped and staff felt the tool was unlikely to raise students' awareness of their energy use on campus. Additionally, the operations team discussed challenges they faced in stakeholder meetings with non-technical university administrators which were often grounded in static reports and spreadsheets. In particular, the energy manager highlighted the challenge of communicating different types of energy data, each with different units, and expressed a desire for visualizations that could communicate multiple variables simultaneously.

After the initial prototype was built, meetings with the university's energy manager, facilities director, and members of the office of sustainability also offered particularly fruitful insights. All staff responded positively to our initial environmental and energy visualizations, and provided detailed feedback which we used to refine the design. Throughout the design process, the initial prototypes functioned as "physical hypotheses" to test the feasibility of our concept and provide direction for future iterations.

## 4.2 Collaborations with Architects

Near the end of the project, we also demonstrated the final system to four architects, who provided feedback about the use of site-specific models and data for public engagement. As with our previous engagements, we began with a demo of the system, then followed a semi-structured interview protocol. We tailored the rest of the conversations based on the background and expertise of each architect, and recorded audio of the conversations which we later transcribed.

## 4.3 Analysis

Throughout the multi-year deployment, we used an ongoing qualitative synthesis approach in which two of the authors regularly reviewed new notes, interview transcripts, and feedback from collaborators. During this process, we maintained and updated a working set of top-level research themes. Over the first two phases, these emergent themes — as well as more specific input from our domain expert collaborators—guided our prototyping efforts and prompted our exploration of the potential for architectural models to support 1) *situating*, 2) *composing*, and 3) *interacting with* geospatial visualizations. In the third phase, we used the results from our interviews with architects and members of the public to refine our higher-level themes as well as identify further opportunities and challenges for integrating visualizations and architectural models. We also used our system as the basis for a small quantitative study, which we describe in section 5.2.

## 5 ARCHITECTURAL MODELS FOR DATA VISUALIZATION

In the following sections, we describe three unique opportunities for integrating physical architectural models and data visualizations on tabletops and illustrate these benefits via our system implementations. Within each section, we critically reflect on these opportunities using feedback and observations from throughout the design process.

### 5.1 Situating Visualizations

The physical characteristics of a scale architectural model can preserve important details about their original referents (including the buildings' size, height, orientation, and layout) that could make it easier to reason about data from them. As such, situating visualizations within and on top of these models can help analysts retain many of the benefits of examining data in the original setting. Moreover, scale models can permit situated analysis and observations from scales and perspectives that are impossible to access in the physical world.

Including the geometric details of the building in terms of height, volume and facade provides valuable information about a particular building such as window and exit locations, which are vital for many types of urban design and architectural analysis [5]. Similarly, the empty space around the 3D model is also representative of places in the real world such as courtyards, parking lots and other spaces that are familiar to the viewer through their experience of the campus. The area around a model surfaces different associations about a space and the buildings within it, and sets up relationships of inside/outside, boundaries, and other spatial relationships. Explorations like Allahverdi et al.'s Landscaper [1] and Buur et al.'s noise curves [4] highlight some of the advantages of incorporating physical representations of data with site-specific physical models. Both highlight the value of maps and models which serve as proxies for locations and make it possible to situate real world data. However, they also emphasize how the lack of depth information in 2D maps can obscure important details that are relevant to analysis.

### 5.2 Recognizability of Maps and Physical Models

To understand how the presence of a physical model might impact viewers' ability to interpret the campus layout, we conducted a between-subjects study in which we asked participants to use either a map or model to identify campus buildings. We recruited 16

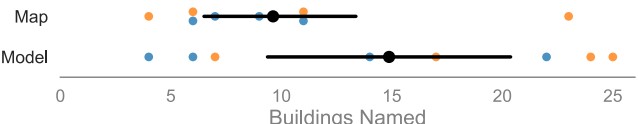

Figure 6: Number of campus buildings correctly identified by participants using only the map (top) and participants using the physical model (bottom). Participants who had spent less than one year on campus appear in blue, while those with more than one year appear in orange. Error bars show 95% bootstrapped confidence intervals.

participants (four female / twelve male, ages 21 to 42) half of whom had spent less than one year on the campus and half of whom had spent at least one year or more. Using either a map with outlines of all buildings on the campus (**map** condition) or the same map projected underneath our physical campus model (**model** condition), we gave participants two minutes to identify and name as many buildings as possible. We provided participants with paper strips listing the names of all campus buildings and asked them to place the names on the tabletop directly on top of the matching buildings. After two minutes had elapsed, a researcher counted the number of correctly-placed names.

The results from our models study with 16 students (Figure 6) suggest that participants who had access to the model tended to be able to more accurately identify campus buildings. While some participants fared poorly in both conditions, only one participant in the map condition was able to correctly identify more than 11 buildings. By contrast, five of the eight participants in the model condition were able to identify 14 or more. Anecdotally, individuals in the model condition reported that they were able to rely on the heights of buildings as well as their visual signatures, allowing them to more readily align their mental model of the campus with the representation on the tabletop.

These findings suggest that physical models can more easily serve as a stand-in for real-world geography, allowing viewers to understand the locations referenced in visualizations and helping them access their own mental model of those spaces, providing context that could help them interpret data. P6 noted that *"the model helped me see what I see everyday"*, while P16 explained that *"without the height I wouldn't have been able to tell which one was MacKimmie Tower"* (a tall landmark building on campus). Another participant, P15, had been on campus for more than two years, and said that the model helped with *"the odd shaped buildings you're used to seeing; that's the tall one, the shape of the buildings helped to see which was which"*.

While our models capture the relative heights and geometry of campus buildings, they still fail to represent much of the finer-grained detail of the buildings themselves, including construction materials, facades, or surrounding greenery. However, our experiences projecting satellite imagery onto the model (as in Figure 1-left) highlight how additional imagery can align well with simple transparent models, providing texture and detail that can give an even richer sense of the real-world environment and further contextualize data.

### 5.3 Composing Visualizations

Any single analysis often involves data from a variety of sources. Visualization designers typically look for ways to join datasets directly using some shared information (an explicit shared key, dates, etc.) in order to visualize them together as a single view. When this is not possible, designers often generate multiple independent visualizations and display them together, using dashboards and overlays to support visual comparison between them.

Spatial and environmental datasets often present a unique challenge, since they frequently use different levels of hierarchical organization which can make it difficult to join datasets directly. Many architecturally-relevant datasets refer to specific point locations (latitude-longitude) or spatio-temporal paths (like the walking trajectories of individuals or vehicles). However, others may refer to regions, buildings, rooms, and other architectural elements with very different scales. This can make it challenging to simultaneously visualize datasets with different scales together (such as building-level energy use and city- or county-level weather data). Moreover, other important pieces of data relevant to the analysis (such as the current price of electricity) may have no spatial component at all.

While most of these datasets can be plotted spatially, simply overlaying them one on top of the other quickly reduces their legibility. We illustrate how designers can use the physical geometry of scale architectural models to compose multiple visualizations together and facilitate transitions between them using the shared context of the model. Specifically, we examine how models and their sub-components can **anchor**, **bound**, and **define the geometry** of visual marks, providing new opportunities for integrating multiple simultaneous views. These approaches allow designers to create composite visualizations [16] that encode more diverse combinations of data, while also creating strong associations between the components of the physical model and the related visualizations, reducing the need for labels and annotations.

**Anchoring.** Using an anchoring approach, the physical positions of an architectural model and its sub-elements define the position of visual marks. Simple examples include positioning visual marks at the centroids of buildings (Figure 7a) or connecting visual marks (or even whole visualizations) to pieces of a model using call-outs or connecting lines. Because anchoring only specifies the position of the visual marks and not their form, it can create a strong visual connection between the visualization and the model while still permitting a wide range of different visual encodings.

**Bounding.** In contrast, a bounding approach uses the shape of a model and/or its sub-elements to separate and contain visualizations. This approach uses the edges and sub-components of the model to divide space to simultaneously show multiple different visualizations both outside (Figure 7b) and inside (Figure 7c). This division of space makes it possible to composite multiple separate visualizations together while creating strong visual associations between visualizations and individual pieces of the model. Bounding can also be used to carve out positive and negative spaces in and around visualizations, creating a stronger sense of alignment between the model and the visualization(s).

**Defining Geometry.** Alternatively, designers can also use the shape of the model to define the geometry of visual marks themselves, creating visualizations that extend the model. For example, colored strokes around the outside (Figure 7d) or inside of a model component can encode categorical or quantitative data related to that element. Similarly, designers can use the geometry of model components as the basis for data-driven shadows (Figure 7e) or extrusions that extend beyond the bounds of the model. While systems like URP [30] and MetaDesk [29] have used these kinds of cast shadows to support light and shadow studies in urban environments, we instead use a shadow metaphor to simultaneously visualize multiple abstract data streams around individual buildings.

### 5.3.1 Prototype Visualizations

Our two example visualizations each use a combination of these operations to create composite visualizations that showcase multiple datasets in and around the model.

**Visualizing campus climate and energy use.** The first visualization (Figure 8-left) uses a **bounding** approach to simultaneously visualize building management data and energy use data for individual campus structures with daily climate data. We created this visualization by integrating daily heating and cooling cost data for individual campus buildings with daily minimum and maximum outdoor temperatures covering a 1-year period from 2016 to 2017. By default, we use the interior of individual buildings to visualize daily heating and cooling costs in that structure, which we encode using a red-blue color ramp. Meanwhile, we use the area around the buildings to visualize a temperature gradient for the same day. Viewers can also toggle the visualization to display electricity and water use inside individual buildings and use a time-series plot below the map to scroll through or play back the entire year's worth of data. Seeing the climate data and energy use together might allow for easy anomaly detection. For instance, viewers can quickly detect if a building is showing high cooling levels even during cold weather events, signalling potential mechanical issues or data quality concerns.

We also examine the use of **anchoring** and **geometry** approaches via a second style of visualization in the *work area* to the right of the main campus map. Here, viewers can examine individual buildings outside of the geographic constraints of the main map, allowing them to display more datasets simultaneously. Here, we encode buildings' overall energy use via the size of a circle anchored at the building's center. Buildings also cast data-driven shadows showing their heating, cooling, electricity, and water use independently.

**Visualizing human movement on campus.** Our second example (Figure 8-right) uses the model to visualize human movement across the campus. We based this visualization on anonymized smartphone location data collected by Galpern et al. in 2017 [11]. This dataset includes 5,530 unique paths drawn from the location histories of 208 students and provides a snapshot of movement patterns across the university over a 4-year period. Because plotting the entire set of paths results in considerable visual clutter, we use the geometry of the model to aggregate and simplify these paths. By default, we **bound** the visualization using the outlines of campus buildings—showing individual paths colored by the movement direction in outdoor areas, but use solid colors to encode aggregated occupancy inside each building. Viewers can also manipulate the visualization to access additional data by interacting directly with the building models.

### 5.4 Manipulating and Authoring Visualizations

Because of their size and shape, architectural models can also serve as graspable tokens, which viewers can use to interactively control visualizations associated with them. Depending on scale and level of detail, models can also be broken down in a variety of different ways, separating pieces into city sectors, blocks, buildings, or even parts of buildings such as floors or staircases to create additional controls. Viewers can then interact directly with the model, moving and arranging pieces to perform a variety of analytic operations.

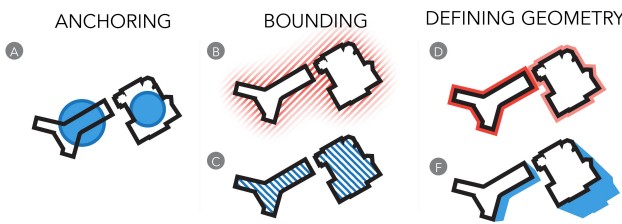

Figure 7: Three approaches for using physical architectural models to compose visualizations. Pieces of a model can (a) anchor visual marks, (b, c) bound and mask visual marks, or dictate the geometry and encodings of visual elements like (d) borders and (e) shadows.

Broadly, tangible interactions allow people to "grasp & manipulate bits by coupling bits with physical objects" [14] and offer a number of benefits, including making user interfaces "more manipulable by using physical artifacts" [9]. Interacting with physical objects can offer a tactile and embodied way of exploring the relationships in complex representations, providing "scaffolds" or cognitive aids that help people solve problems that would be more difficult using "brain-internal computation" [6]. Moreover, tangible interaction can be a valuable tool for embodied sensemaking [13].

The use of tangibles on tabletops has been widely explored in other domains, but presents a particular set of challenges and opportunities for architectural models. On one hand, architectural tabletop models are a core component of architectural design practice, where scale models are still routinely crafted and manipulated by hand and serve as a locus of design exploration [12]. However, in contrast to other instances of tangibles on tabletops, architectural models as input devices have limited degrees of freedom—constrained by the physical characteristics of the models themselves. (For example, most building models have a natural up and down, and thus are unlikely to support rotation around multiple axes.) Despite these constraints, physical models offer a rich set of possible interactions via which viewers can reconfigure models to gain new information, while simultaneously leveraging the recognizable form and physical properties of the pieces themselves.

Based on these insights, we used our two prototypes to examine four specific interaction techniques (Figure 9) which use buildings as physical interaction tokens. These include several interactions in which viewers interact with models to alter visualizations while preserving the original spatial layout. We also showcase how models can support grouping, reorganizing, and re-configuring visualizations when these spatial constraints are relaxed.

**Reveal.** In a *reveal* interaction, picking up a piece of the model can be used to hide or show information in the visualization. These simple interactions can work well when models are placed in a fixed geospatial layout (like the campus map) and translating or rotating them would disrupt that configuration. In these cases, reveal interactions can trigger queries and filters or change the properties of the underlying visualizations that do not impact their layout. For example, our movement visualization introduces a reveal interaction (Figure 9a) in which lifting a building off of the tabletop hides the occupancy data for that building and reveals the raw movement paths underneath. This particular interaction builds on the intuition of lifting a physical object to reveal the area or objects beneath it.

**Assemble.** Conversely, in an *assemble* interaction, repositioning pieces of the model on the map serves as a mechanism for constructing new visualizations that selectively reveal information associated with individual pieces while still retaining a fixed spatial layout. We explore this concept in our movement visualization by allowing viewers to clear the tabletop of all models, then selectively re-add buildings to reveal only the paths that pass through all of them (Figure 9b). These new views make it possible to examine distinct subsets of the data, letting viewers examine specific flow paths and bottlenecks on campus, while reducing clutter both on the tabletop and in the visualization.

**Extract.** In an *extract* interaction, pieces of the model can be repositioned to create new visualizations which ignore the spatial constraints of the original model, allowing viewers to create dramatically different visual configurations. By extracting models from their geospatial context and placing them into a more flexible space, viewers can surface additional information that might be hidden or occluded in a spatial layout. In both of our example visualizations, we support these interactions by including a *work area* to the right of the main map. In the movement visualization, we use this extract interaction to display building names and encode information about the number of paths that pass through the building. Meanwhile, in the climate visualization, placing objects into the work area reveals data-driven shadows which simultaneously reveal information about that building's heating, cooling, electricity, and water use (Figure 9c). Viewers can also re-position buildings in the work area to create clusters, orderings, and other layouts. Because the models retain the recognizable form of the original building, they remain easy to identify and reason about even when removed from their original geospatial locations. As in prior systems like reacTable [17], tangible models could also be used to dynamically construct new kinds of charts, including network visualizations.

**Reorient.** Similarly, in a *reorient* interaction, pieces of the model can be rotated on the tabletop to provide additional input to the system. While this may be impossible in geospatial layouts where pieces appear close together, the rotation of pieces in non-spatial layouts can provide a rich, continuous input mechanism associated directly with a specific building. In our movement visualization, we examine how these rotation interactions could be used to filter the underlying data based on direction of travel (Figure 9d). Viewers can rotate models that they have placed in the work area like dials. These buildings then serve as simple angular selection widgets [25] that filter the main visualization to show only the traffic that passes through that building in a specific range of orientations.

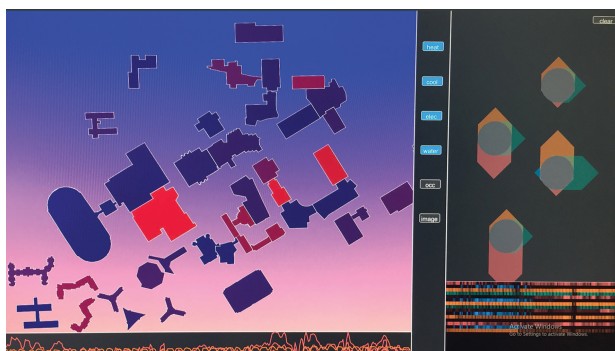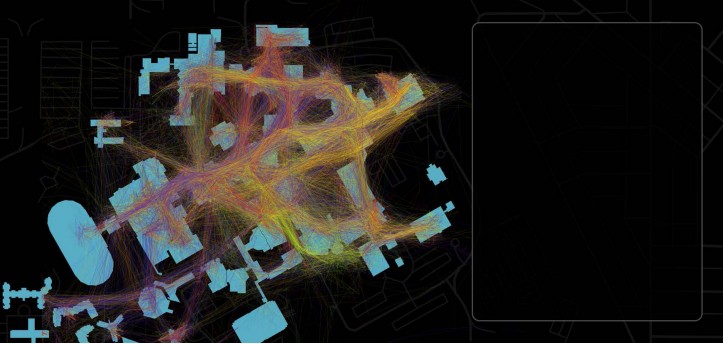

Figure 8: Screenshots of our visualizations without the physical model. The campus climate visualization (left) showing daily heating (red) and cooling energy (blue) for individual campus buildings with that day's temperature gradient in the background. The work area at right shows heating (red), cooling (blue), water (green), and electrical (orange) usage for specific buildings. The movement visualization (right) overlays movement paths with occupancy data from inside the buildings—shown here with paths visible both inside and outside of buildings and direction of travel encoded using color.

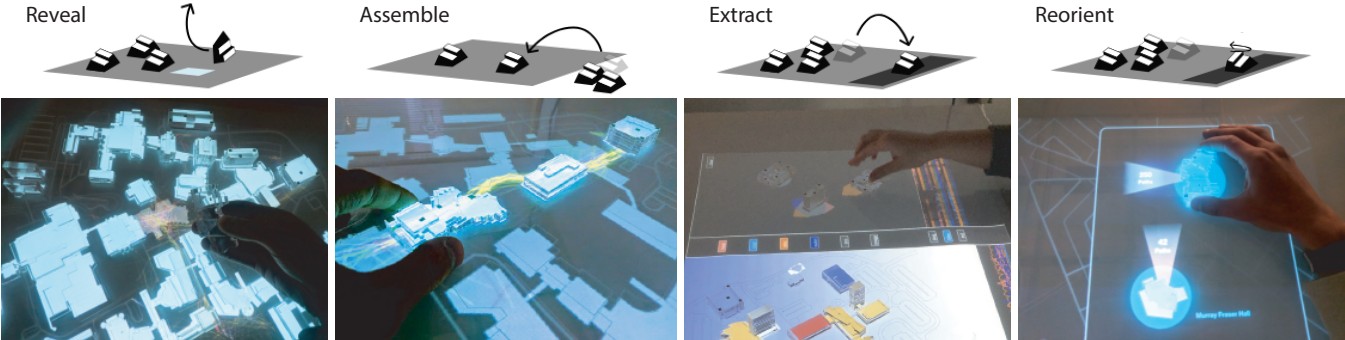

Figure 9: Four interaction techniques (reveal, assemble, extract, and reorient) that use manipulation of physical models to interactively control the layout and detail of the visualizations.

## 6 DISCUSSION

Our ongoing process of design, reflection, consultation, and validation surfaced a variety of implications for future systems that integrate visualizations and architectural models. In our discussion, we offer four takeaways for layering multiple data sets with physical model. While we have discussed the potential for simultaneous worlds in the context of architectural models, we also highlight how the concept may hold promise in other application domains.

### 6.1 Models and Data Granularity

Physical models are static objects limited to the scale at which they are fabricated, which in turn limits the scale and granularity of the data visualizations they anchor. This can present challenges if the level of detail of the data and visualization are not compatible.

In our energy visualizations we map energy use to the color of the building outline on a building-by-building basis. This suits the granularity of the current data, which is monitored by a single meter in each building. Both the energy managers and sustainability directors preferred this scale because it matched the scale they used to analyze the university systems. They also emphasized the value of seeing energy use data together with a building's facade and volume, since this combination reveals relationships between energy use and size that are not visible in spreadsheet data—with the energy manager emphasizing, "we never see the data this way, it brings up other things to think about."

The architects, however, questioned the usefulness of visualizing data for entire buildings, as they felt that seeing information on a room-by-room basis would be more likely to show inefficiencies and other issues. This way, operators who are used to a "normal pattern of usage" for each building could see outliers or anomalies within each building. Additionally, one architect suggested that the system might be used as a control panel in other applications, for example for airport security and logistics — again suggesting that showing data on a floor-by-floor or room-by-room basis might be more useful for detecting patterns.

However, using models as a substrate for more granular data poses several practical challenges. For example, surfacing more granular data on a campus-scale model composed of individual buildings would likely reduce designers' ability to use models as interactive controls, since each building would now visualize multiple data points. However, breaking models into smaller pieces make them more difficult to manipulate and to track. In these cases, designers may need to explore hybrid solutions that use models to show data at one level of abstraction but support exploration of more granular data using other representations. For example a tabletop system like ours might show building-level aggregates on the model itself, while displaying floor-by-floor or room-by-room data in the work area next to the model.

### 6.2 Optimizing City-Scale Systems

Campus, city, and neighborhood-scale architectural models present opportunities for understanding and evaluating larger meta-systems together with their component pieces. Interactive and modular physical models can help facilitate this interplay—but their physical nature limits the potential for analyses that bridge more distant scales.

In our discussions, the campus operations manager appreciated the combined visualizations with multiple types of data at one time, noting that "we can see the sustainability of the campus as a whole, which compliments and expands our understanding of energy use for whole building optimization, which considers energy management of the campus as a whole." Going forward, the goal of the university is to identify excess energy and store it for future use or transfer to other areas nearby when required. For example, if a building such as an ice skating arena is producing excess heat, the operations department might store or transfer the heat to use in other nearby buildings.

Integrated visualizations like our climate tool could make it easy to monitor where on campus excess cooling or heating is occurring, and which buildings close by are in need of that type of energy. The facilities manager noted that the geospatial layout made it easy to see the energy loop from the plant to each building and back again. These considerations are especially important when evaluating potential building upgrades, where even incremental reductions in energy use in large, well-designed buildings can translate into substantial overall savings. Like the architect's considerations for more granularity, the operations manager considered aspects on a different scale than we designed our model for. The operations manager needed to keep an eye on the big picture. Thus, they were more inclined to consider campus design as a system. In terms of using architectural models, this suggests considering ways that multiple pieces of a model might connect physically to be considered as one.

### 6.3 Collaboration around Scale Models

The use of tabletop tools as collaboration platforms is well-documented [26,27] and our feedback supports these findings. Moreover, our interviews with stakeholders and the response to our public demos suggests that situating visualizations using a model provides a strong and engaging entry point and encourages viewers to treat the tabletop as a collaborative tool. Additionally, the administrators we worked with felt that the combination of visualization and model would make it easier for non-technical users to understand the relationship between buildings' form and their energy use. Administrators also highlighted the potential for using the model as a "control center" on which to visualize diverse situational and operational data. They noted that such a display could be used by multiple departments on campus to help develop a better understanding of their operations and thus aid interdepartmental meetings. Similarly, the operations and sustainability managers emphasized how this

kind of model could help frame public discussions around proposed buildings and retrofits, allowing stakeholders to more readily appreciate both the physical interplay of buildings and their relative energy footprints.

## 6.4 Using Scale Models for Exploration and Simulation

Our prototypes all used detailed models of the current campus to visualize historical and real-time data. However, our discussions with architects highlighted the potential for models to support the exploration of predictive simulations and future scenarios, and we consider this a particularly rich area for future work. In particular, one of the architects surfaced an essential distinction between models that reflect the current state of the world and those that embody alternative possibilities. Currently, most tangible tabletop systems designed for planning or educational purposes [27] represent abstract scenarios using generic or primitive tangible forms. These abstract scenarios can allow participants to model and imagine many different potential campus designs free from the constraints of the current configuration. However, energy managers and other administrators are interested in how the model reflects the campus operationally, as it is in real life.

Considering how to support both monitoring and simulation scenarios is an ongoing challenge with numerous trade-offs. For example, representing buildings using generic geometric primitives frees viewers to imagine a variety of possible design alternatives, but may miss opportunities for situating predictive simulations in an accurate architectural context. Going forward, modular, reconfigurable, or shape-changing models have the potential to address these challenges, allowing the same models to transition between abstract and realistic forms as needed. This suggests the next exciting steps for integrating architectural models and tabletop user interfaces might look to recent work in shape-changing interfaces [23]. We encourage future work to understand how we might build models that integrate possibilities for simulating "what might be".

## 7 CONCLUSION

We present a design exploration examining how physical architectural models on tabletops can anchor visualizations of multiple complementary datasets. Our explorations detail how physical models offer new potential for supporting observations that integrate contextual and ambient information from multiple "Simultaneous Worlds". Specifically, our work highlights how physical models can help situate and compose multiple visualizations together, while also serving as tangible tokens that allow viewers to manipulate and author new representations. The layering of heterogeneous data streams around a physical model creates new opportunities for situated data analysis, which we are actively exploring in our continuing research. Through both our design explorations and reflections on the development process, we hope to lay the groundwork for even richer integration between visualizations, physical architectural models, and interactions, including ones that extend beyond the context of tabletops. Moreover, we hope this work provides inspiration for other forms of physical data models that support situated and embedded visualization with embodied and fluid interactions.

## ACKNOWLEDGMENTS

Thanks to Kevin Ta, Max Kurapov and Kurtis Danyluk for their valuable assistance with the project. This work was supported in part by Alberta Innovates Technology Futures (AITF), the Natural Sciences and Engineering Research Council of Canada (NSERC) [RGPIN-2016-04564], the Canada Research Chairs Program, SMART Technologies, the University of Calgary SU Sustainability Fund and GSA Quality Money Grant, the Canada Foundation for Innovation, and the European Union's Horizon 2020 research and innovation programme under the Marie Sklodowska-Curie grant agreement No. 753816.

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
