# OpenReview forum: "Simultaneous Worlds: Supporting Fluid Exploration of Multiple Data Sets via Physical Models"
_graphicsinterface.org/Graphics_Interface/2022/Conference — GI 2022_

### Official Review · Reviewer_Fi5k · 2022-04-11
**Study of scale architectural models as a way to situate, compose, and interact with data visualizations. Valuable insights in tangible interaction capabilities and documentation of design process.**

**Rating:** 7
**Confidence:** 4

**Review:**

==Originality:==
This paper presents an in-depth study of using scale architectural models for data visualization. There are three main contributions:
- Showcasing methods to situate and compose data visualizations in the context of architectural models. The authors introduce anchoring, bounding, and geometry definition as three approaches to data plotting.
- Interactively manipulating and authoring data visualizations by using scale models as graspable tokens. They introduce four approaches including revealing, assembling, extracting, and re-orienting visual data as a result of the tangible interactions.
- Documentation of the design process including feedback from domain experts over a 2 year period.

Despite there being ample previous work on tangible tabletop interaction, the paper provides valuable insights on data visualization specific to campus/urban models. In my view, the tangible tokens idea presented in sec. 5.4 is the most compelling and should be given more emphasis. The interaction approaches (reveal, assemble, etc.) appear to be natural and effective ways of leveraging the physical building models, and allowing users to explore data sets in a geospatial setting. The ideas for anchoring, bounding, and defining the geometry (sec. 5.3) are also insightful, but would also be possible in a standard 2D setting.

==Clarity:==
The paper is clearly written and enjoyable to read. However, the contributions could be more clearly expressed:
- I'd suggest removing "simultaneous worlds" from the title since this aspect was less pronounced in the results. Only 2 datasets were overlaid in the examples.
- Fig 2 suggests that many visualization layers will be simultaneously displayed. But this is not reflected in the examples shown later. If there is another example that could be added, that would help strengthen the simultaneity contribution, otherwise I think that aspect should be downplayed.

Other minor points on writing:
- In sec. 4 it would be helpful to have subsection titles that match up with the phases in Fig. 5: design phase, 1st prototype, 2nd prototype, reflection. This would improve the reader's understanding of the timeline.
- I'm confused about the building occupancy visualization in Figs. 8, 9. The buildings all seem to be the same blue color, how is occupancy being shown?

==Validation:==
Two detailed visualizations are demonstrated: energy use and climate in example 1, and movement data and building occupancy in example 2. If available, it would be helpful to see more examples with a higher number of datasets since the emphasis is on examining multiple streams of geospatial data. This paper would also benefit greatly from a supplemental video to demo the setup and example visualizations.

Feedback from domain experts is also well documented, listing feedback at various points in the design process (e.g. at the ideation stage, after the initial prototype, and for final reflection). It would be helpful to hear how often experts were consulted over the two year period (paper just says "periodic")

Regarding the physical model, did any experts comment on the tracking markers interfering with the data visualization? Similarly, were there any issues with the acrylic models distorting the view of the projected data, in particular, detailed information like movement paths?

---

### Official Review · Reviewer_rwQF · 2022-04-13
**Simultaneous Worlds: Supporting Fluid Exploration of Multiple Data Sets via Physical Models**

**Rating:** 5
**Confidence:** 2

**Review:**

Summary:

The authors present their work on designing a table top visualization tool to display data for a university campus -- energy usage data,
 student travelling data, etc. They use a bottom projected display and track miniature buildings using computer vision techniques. They can remove, replace, reorient buildings with their system to interact with the display. The authors consulted architects and campus building operations to gather feedback on the value of their system. They also performed a user student to demonstrate that miniature buildings provided better visual aids than simple building outlines.

Feedback:

The writing style of the paper could use improvement. There are many instances of vague language that doesn't really communicate anything to the reader. For example, the paragraph at the end of page 1 could be replaced with a single sentence that says "We iterated on our design based on feedback from experts in building operations and architecture". The following paragraph suffers from the same issue and many other places throughout the paper. Please try to be as specific, concrete and concise as possible. We a reading your paper to learn your brilliant ideas, try to make it easy on us by telling us directly what you want us to know. I would encourage the authors to read through a textbook on technical and persuasive writing.

The exposition could also be tightened up, the technical details of the system are peppered throughout the paper, between repeated references to consulting building ops and architects. The paper could be re-organized with the technical details of the display system, the design feedback and the user studies as self contained pieces. For example, I read about the tracking system in 3.3 but I didn't understand why we needed to track buildings until 5.4.

Finally, as a technical contribution, the paper isn't very convincing that the inflexibility of using miniature is worth the trade off of a simple touch screen table display. The user study for identifying buildings was a neat idea, but can't we just put the names of the buildings on the display? Because the paper is focused on why it's important to have a table top design with miniatures, it ought to really emphasize why only a table top with miniatures can achieve the goals of the project.

This is a difficult paper to judge. It's hard to distill down exactly what are the research contributions and exactly why this paper is valuable to the research community. I would recommend a resubmission after a thorough edit with technical and persuasive writing in mind. If other reviewers find it valuable, I am also okay with accepting.

---

### Official Review · Reviewer_qkyG · 2022-04-14
**Well motivated and well excecuted design exploration**

**Rating:** 9
**Confidence:** 4

**Review:**

This paper presents a design exploration using tangible building models for data visualisation. Inspired by the use of tangible models in architecture, the paper explores how such models can be used to support understanding and exploration of different types of data through visualisations situated on a campus map. The work evolves through several phases of discussion with domain experts in facilities management and energy. These discussion lead to the exploration using models to support three different categories of use (situating, composing, and interaction). Two prototypes demonstrate these uses along with several interesting ways of utilisting the model geometry in the data visualisation, and of manipulating the models during interactive exploration.

The paper is very well written, organised and interesting to read. The work is well-motivated in drawing from architecture and well-grounded in prior work. The discussions are insightful and well supported by the use of figures. The system implementation introduces several interesting and novel uses combining tangible interaction and data visuslisation. Overall this is high quality work that provides an interesting contribution to the data vis community.

Additional comments:
- In the discussion of building heights, I see the motivation for including scaled heights, but is this a scalable approach? For instance would this work in a visualisation of a typical downtown core with a mix of skyscrapers and much shorter buildings?
- I followed the concept of using shadows for data representation (5.3.1), but didn't exactly how this is applied in Figure 8. I get the central mark (circle), but wasn't sure how to interpret the other shapes.

---

### Decision · Program_Chairs · 2022-04-17

Accept